# The Dynamic Conversion of Dietary Protein and Amino Acids into Chicken-Meat Protein

**DOI:** 10.3390/ani11082288

**Published:** 2021-08-03

**Authors:** Shemil P. Macelline, Peter V. Chrystal, Sonia Y. Liu, Peter H. Selle

**Affiliations:** 1Poultry Research Foundation, Department of Animal Science, The University of Sydney, Camden, NSW 2570, Australia; shemil.macelline@sydney.edu.au (S.P.M.); pchr7039@uni.sydney.edu.au (P.V.C.); sonia.liu@sydney.edu.au (S.Y.L.); 2School of Life and Environmental Sciences, Faculty of Science, The University of Sydney, Camden, NSW 2570, Australia; 3Sydney School of Veterinary Science, The University of Sydney, Camden, NSW 2570, Australia

**Keywords:** amino acids, broiler chickens, glucose, protein, starch

## Abstract

**Simple Summary:**

The conversion of dietary protein and amino acids into chicken-meat is a dynamic and complex process. Dietary protein is transferred to protein in a chicken carcass at a conversion ratio in the order of 2.50:1, which leaves scope for improvement. Nevertheless, this conversion ratio in broiler chickens cannot be matched by other terrestrial food-producing animals. The quest for sustainable chicken-meat production would be greatly facilitated by enhancing the efficiency of this conversion. Therefore, this review explores the various pathways and processes involved with the objective of identifying approaches and strategies whereby the transition from dietary protein to chicken-meat protein can be advanced.

**Abstract:**

This review considers the conversion of dietary protein and amino acids into chicken-meat protein and seeks to identify strategies whereby this transition may be enhanced. Viable alternatives to soybean meal would be advantageous but the increasing availability of non-bound amino acids is providing the opportunity to develop reduced-crude protein (CP) diets, to promote the sustainability of the chicken-meat industry and is the focus of this review. Digestion of protein and intestinal uptakes of amino acids is critical to broiler growth performance. However, the transition of amino acids across enterocytes of the gut mucosa is complicated by their entry into either anabolic or catabolic pathways, which reduces their post-enteral availability. Both amino acids and glucose are catabolised in enterocytes to meet the energy needs of the gut. Therefore, starch and protein digestive dynamics and the possible manipulation of this ‘catabolic ratio’ assume importance. Finally, net deposition of protein in skeletal muscle is governed by the synchronised availability of amino acids and glucose at sites of protein deposition. There is a real need for more fundamental and applied research targeting areas where our knowledge is lacking relative to other animal species to enhance the conversion of dietary protein and amino acids into chicken-meat protein.

## 1. Introduction

The conversion of dietary protein and amino acids into the protein of chicken-meat is indeed a dynamic process. Quite typically, broiler chickens attain a live weight of 2.918 kg at 42 days post-hatch and a carcass weight of 2.151 kg following processing. This translates to 376 g of carcass protein as a Ross 308 broiler carcass contains 175 g/kg protein [1]. Broiler chickens consume 4.702 kg of feed over 42 days with dietary protein contents declining from 230 to 183 g/kg, with a weighted average of 201 g/kg protein. This corresponds to an intake of 945 g protein for an output of 376 g. Therefore, 2.51 kg of dietary protein is required to generate 1.00 kg of protein in a chicken carcass or saleable end product. The dietary protein to carcass protein ratio of 2.51 is unmatched by other terrestrial food-producing animals. The efficiency of protein gain in broiler chickens (33.3%) was estimated to exceed that of pigs (23.3%) and feedlot cattle (12.1%) by clear margins [2]. Nevertheless, if reduced-crude protein (CP) diets could be developed so that a dietary reduction of 50 g/kg CP did not compromise growth performance, then the dietary protein to carcass protein ratio of 2.51 would decline to 1.89, a marked 24.7% improvement. The overall efficiency with which dietary amino acids are incorporated into chicken-meat, based on the data of He et al. [3], is in the order of 55%, but this may vary from 35% to 70% for a given amino acid. Thus, this review considers the transition from dietary protein and amino acids to carcass protein in broiler chickens with the intention of identifying strategies to enhance the complex, kinetic processes involved.

Soybean meal is the dominant source of dietary protein for chicken-meat production, supplying up to 70% of protein in a typical ‘corn-soy’ diet while the balance is derived from feed grains and feed-grade amino acids, usually lysine, methionine and threonine. While there are alternatives to soybean meal, the US chicken-meat industry utilised 15 million tonnes of soybean meal in 2017–2018, which is 48% of the total usage in that country [4]. Global production of soybean meal was 360 million tonnes in 2018/2019 [5] and chicken-meat production uses substantial amounts of this production. A decade ago, it was estimated that 11.4 million tonnes, or 32% of the total of 35.8 million tonnes, of soybean meal was fed to broiler chickens in the USA [6]. Global soybean production is forecast to increase from 369 million tonnes in 2020 to 407 million tonnes in 2027, and the price of soybeans is predicted to increase from USD 416 to USD 453 over the same timeframe [7]. The likelihood is that the price and supply situation of soybean meal will become an increasingly real challenge to sustainable chicken-meat production given the FAO projected robust growth in demand for chicken-meat to 2050 [8].

Consequently, there is an urgent need to develop alternative dietary protein sources to partially replace soybean meal, which could range from meals based on black fly soldier larvae [9] to enhanced canola meal [10] in addition to reduced-CP diets. Moreover, a reduced demand for soybean meal by the chicken-meat industry would attenuate neo-tropical deforestation in South America [11]. Synthetic and crystalline, or non-bound, amino acids were considered as alternatives to soybean meal, as reviewed by Selle et al. [12]. These researchers contended that the chicken-meat industry’s dependence on soybean meal would be halved if CP reductions of 50 g/kg are realised without compromised broiler growth performance. This would involve judicious dietary inclusions of non-bound amino acids to meet requirements. There are copious inclusions of non-bound amino acids in reduced-CP diets, thus the question is raised as to whether the bioequivalence of non-bound and protein-bound amino acids are identical. This appears unlikely given the different rates of intestinal uptakes between the two entities [13], which is addressed in this review. Finally, that broiler chickens achieve live weights of 3 kg in six weeks represents rapid growth rates and emphasises the need to consider the kinetics of dietary protein and amino acids transitioning to protein in chicken-meat, which is entwined with the digestive dynamics of starch and protein.

Thus, the objective of this review is to explore the transition of dietary protein and amino acids into carcass protein in broiler chickens and to identify strategies that may enhance this conversion. That this is discussed in relation to other animal species, including pigs, rats and humans, is indicative of the lack of data specific to avian species in some key areas. 

## 2. Dietary Sources of Protein and Amino Acids

Diets for broiler chickens are usually formulated on a least-cost basis to meet specified targets for a selected range of digestible amino acids and energy density. Numerous amino acid profiles have been recommended so that the ‘ideal protein ratio’ may be achieved. Ten examples of recommended ideal protein ratios are shown in Table 1 and they merit careful perusal because variations in these recommendations have been quite substantial over the decades. For example, the mean ratio of histidine relative to lysine is 34.4 ± 5.27 with a minimum of 27 and a maximum of 44. Three studies [14,15,16] have been identified in which ideal protein ratios have been compared and all three studies recorded tangible differences in growth performance in response to the various ideal protein ratios investigated. Instructively, Macelline et al. [17] compared two sets of ideal protein ratios in broilers offered wheat-based diets with CP contents of either 210 or 180 g/kg from 14 to 35 days post-hatch. ‘Ratio A’ was the recommendation of an amino acid supplier and ‘Ratio B’ was that of an American University. ‘Ratio B’ was significantly superior in 210 g/kg CP diets generating an improvement of 4.17% (1.424 versus 1.486) in FCR, whereas ‘Ratio A’ supported a significant FCR improvement of 3.19% (1.580 versus 1.632) in 180 g/kg CP diets. This outcome is a clear indication that an ideal protein ratio for a reduced-CP diet is almost certainly different from that of a standard diet.

Average amino acid compositions of six samples of relevant feedstuffs were determined by Li et al. [28]. The amino acid concentrations ratios of soybean meal, fishmeal, meat and bone meal, maize and sorghum are compared with the Texas A&M ratios in Table 2. The Texas A&M University’s optimal ratios of true digestible amino acids, relative to lysine (100), as proposed by Wu [27], are the most extensive set of recommendations as they take the so-called non-essential amino acids into consideration. Interestingly, the linear relationships between amino acid ratios in feedstuffs with Texas A&M ratios show that fish meal (*r* = 0.920; *p* < 0.0001) is the most closely aligned protein source, followed by meat and bone meal (*r* = 0.873; *p* < 0.0001), soybean meal (*r* = 0.770; *p* < 0.0001), sorghum (*r* = 0.692; *p* < 0.001) and maize (*r* = 0.657; *p* < 0.005). Animal protein sources are usually considered superior to vegetable proteins because their higher leucine, lysine and methionine concentrations may have anabolic effects [29]. However, usage of animal proteins is restricted in non-ruminant diets by their prohibition in the European Union and other countries, and their prices may not be competitive due to demands from aquaculture and the pet food industry. Inclusions of meat and bone meal and other animal proteins in diets for food-producing animals were banned in Europe, as a precaution, due to concerns over the possible transmission of Bovine Spongiform Encephalopathy (‘Mad Cow’ Disease) to humans. However, the protein contents and digestibility of amino acids in meat and bone meals are highly variable, which is influenced by numerous factors, including rendering conditions and relative CP contents to calcium and phosphorus contents [30]. Limitations in plant proteins include the presence of antinutritional factors such as phytic acid or phytate, non-starch polysaccharides, trypsin and amylase inhibitors, phenolic compounds, possibly condensed tannin and saponins [31].

The digestibility of protein and amino acids in relevant feedstuffs in broiler chickens has been extensively documented. Apparent and standardised ileal digestibility coefficients of amino acids in feedstuffs have been documented [32,33,34,35,36], as listed by Ravindran et al. [28]. A collaborative study was reported by Ravindran et al. [37], in which apparent ileal amino acid digestibility coefficients of the same maize-soy diet were determined by five institutions using common assay procedures. The outcomes are shown in Table 3. Arginine (0.912) and methionine (0.905) were the most digestible essential amino acids and the threonine (0.800) the least. Digestibilities of non-essential amino acids ranged from glutamic acid (0.902) to cysteine (0.758). 

Maize and wheat are the commonly used feed grains in broiler diets globally. Table 4 provides the composition of maize- and wheat-based diets with CP concentrations of 222 and 165 g/kg that were offered to birds from 7 to 35 days post-hatch in [38]. The reduction in dietary CP from 222 and 165 g/kg generated, on average, an increase in non-bound amino acids by a factor of 5.97, an increase in protein derived from either feed grain by a factor of 1.42 and a 74.6% reduction in soy protein. However, these marked shifts in protein derivations are noticeably amplified in wheat-based diets because of the higher protein content in wheat. In theory, these differences are accommodated by the formulation of diets based on digestible amino acids. However, this assumes that the digestibility of non-bound amino acids is 100%, which is generally accepted [39], and that the digestibility of amino acids from different sources are additive. On average there is a 42% increase, from 518 to 736 g/kg, in feed grain inclusions pursuant to the dietary CP reductions in Table 4, and a similar increase in starch concentrations. This raises the possibility that additional dietary starch may interfere with protein digestion, which could stem from competition for intestinal uptakes between glucose and amino acids [40,41,42]. This possibility is considered in more detail further on in this review. 

## 3. Digestion of Protein and Intestinal Uptakes of Amino Acids 

Enzymatic digestion of protein by endogenous proteolytic enzymes and intestinal uptakes of amino acids along the digestive tract is reasonably well understood and a thorough review of protein digestion in poultry has been documented by Moran [43]. Additional reviews of protein digestion in poultry and livestock have been published by Krehbiel and Matthews [44], and in poultry specifically by He et al. [3], while an instructive paper focusing on amino acid absorption in humans is recommended [45]. Importantly, the digestion and absorption of nutrients is an energy demanding process accounting for perhaps more than 20% of dietary energy [46]. 

The protein digestive process is initiated in the proventriculus of poultry where low pH generated by HCl secretions denatures proteins and facilitates their cleavage by pepsin. Peptide end products of pepsin digestion trigger the release of CCK and gastrin, which probably play a regulatory role in the overall protein digestive process [44]. Several proteolytic enzymes originating from the pancreas (trypsin, chymotrypsin, carboxypeptidase, elastase) are released into the duodenum to convert polypeptides into short peptide fragments. These are, in turn, converted to di- and tripeptides (or oligopeptides) by amino peptidase and dipeptidase on the apical membrane of enterocytes. Intestinal uptakes of di- and tripeptides take place via the oligopeptide transporter, PepT-1 [47,48,49]. Alternatively single, monomeric or non-bound amino acids are absorbed via an array of Na^+^-dependent and Na^+^-independent transport systems with overlapping specificities and differing affinities [50,51]. Na^+^-dependent transporters are probably the more dominant and co-absorb amino acids and sodium (Na) in response to Na-pump activity in the basolateral membrane of enterocytes. In contrast, PepT-1 is not directly Na^+^-dependent and the distinction between monomeric amino acids and oligopeptides is almost certainly important.

The assertion of Krehbiel and Matthews [44] was that 70 to 85% of amino acids are absorbed as oligopeptides as opposed to monomeric amino acids. The intestinal absorption of peptides has been considered extensively by Matthews [52,53] and his colleagues. For example, a partial hydrolysate of lactalbumin was compared with an equivalent blend of free amino acids in humans [54]. The proportion of α-NH_2_N absorbed from lactalbumin hydrolysate perfusions was 25.5 percentage units (56.6 versus 31.1%) greater than from the free amino acid mixture. Moreover, it has been asserted that intestinal oligopeptide uptakes are more rapid and efficient than monomeric amino acids [55,56]. This may be the case, however, polypeptides must be digested to oligopeptides before this advantage is declared. Therefore, the salient point is that reduced-CP diets contain less intact protein and, in turn, less oligopeptides and more non-bound amino acids than standard-CP diets, which may disadvantage reduced-CP diets in respect of intestinal uptakes of amino acids. Importantly, intestinal uptakes of amino acids are probably more limiting on broiler growth performance than the digestion of protein [57].

The retention time, pH of digesta, activities of trypsin, chymotrypsin, amylase and lipase in individual segments along the digestive tract of broiler chickens are shown in Table 5. The pH of digesta was determined in birds offered diets containing 10.7 g/kg calcium [58] and retention times were determined with ^103^ruthenium phenanthroline [59]. It is noteworthy that digesta is retained for only 59.5 min in the jejunum (17.6% of total retention time of 338 min) and yet intestinal uptakes of nutrients mainly take place in this segment. Accordingly, digestive enzyme activities are noticeably higher in the jejunum [60]. However, digestion retention times along the avian gastrointestinal tract are not as straightforward as Table 5 may imply. 

Retention of digesta in the crop and gizzard is tabulated at 7.4 and 50.2 min, respectively, but actual digesta retention times are subject to variation in both these organs. Moreover, digesta retention times along the entire digestive tract are complicated by episodes of reverse peristalsis, particularly the recycling of digesta from the duodenum into the gizzard [61]. Broiler chickens have the capacity to store large quantities of feed in their crops [62] and, in anticipation of scotoperiods or periods of darkness, birds will ‘crop-up’ and retain digesta in the crop [63]. This was demonstrated by Yin et al. [64] as birds with 12 h access to feed had relative crop weights of 7.4 g/kg, as opposed to 3.8 g/kg in birds with 20 h access where the corresponding relative weights of crop contents were 35.4 and 0.8 g/kg. Reverse peristalsis includes the retrograde movement of digesta between the proventriculus and the gizzard, the small intestine and the gizzard, the rectum and the small intestine and the cloaca and the rectum [65]. Interestingly, the gizzard is believed to be the pacemaker of gut motility and regulates episodes of reverse peristalsis [66]. 

Thus, the retention of digesta in the crop and gizzard, which is believed to be amplified by whole grain feeding, and the recycling of digesta along the gastrointestinal tract via reverse peristalsis clearly differentiates avian from mammalian species. Inert dietary markers, including titanium oxide and acid insoluble ash, are routinely used to determine amino acid digestibilities in birds. However, given the retention and recycling of digesta, there is the possibility of discrepancies in the passage rates of marker and all components of digesta. Moreover, the passage rate of solids through the gizzard is 40 min as opposed to 17 min for liquids, as discussed by Sklan et al. [67]. This raises the possibility that soluble, non-bound amino acids with low molecular weights, may separate or leach out in the liquid phase of digesta retained in the crop and gizzard and may not be recycled equally by reverse peristalsis. These differences may even partially explain why pigs [68] are better able to accommodate reduced-CP diets than poultry [69], as is evident in these two publications.

### 3.1. Digestibility Coefficients of Amino Acids

The standard procedure is to determine apparent amino acid digestibility coefficients in digesta taken from the terminal ileum. While this eliminates the confounding effects of hindgut fermentation it overlooks the fact that protein digestion and absorption of amino acids takes place mainly in the jejunum. Amino acids that remain present in ileal digesta may be derived from the diet, from endogenous secretions or from gut microbiota. Additionally, standardised or true digestibility coefficients of amino acids may be assessed. Standardised digestibility is determined by correcting apparent digestibility values for basal endogenous losses and true digestibility is determined by correcting for both basal and specific endogenous losses [70,71]. However, the accuracy with which basal and specific endogenous losses can be determined is somewhat problematical.

Inevitably, there are endogenous amino acid flows in broiler chickens comprised of digestive enzymes, mucin and desquamated intestinal epithelial cells augmented by amino acids from microbiota, which are not really of endogenous origin, as reviewed by Ravindran [72]. Mucin, which in pigs has a protein concentration of 343 g/kg, is an important source of endogenous amino acids because mucin protein is essentially not digested and its amino acids are not re-absorbed [73,74]. The sequence of amino acids of selected endogenous secretions, including avian mucin, and of gut microbiota is shown in Table 6. Pearson correlations between amino acid sequences of mucin, pepsin, trypsin, amylase, microbial and whole-body protein are presented in Table 7. Interestingly, the amino acid sequence of whole-body protein is significantly correlated with trypsin, amylase and microbial protein, not with mucin and pepsin, but the extent to which amino acid contents from one source are related to others is evident.

Consequently, identifying the constituency of the pool of amino acids remaining in ileal digesta is a challenge. However, Duvaux et al. [81] developed a mathematical method to approximate proportions of different protein sources in a mixture based on their sequence of amino acids. The proportional origins of amino acids remaining in distal ileal digesta from three studies completed in this institution were subjected to the [81] model, as shown in Table 8. Amino acids of dietary origin comprised an average of 52.7% of the total, those of endogenous origin 26.4% and 20.9% of amino acids were of microbial origin. Using the same mathematical model, Le Guen et al. [82] estimated that the origin of amino acids in ileal digesta comprised 25% dietary, 30% endogenous and 45% microbial in pigs offered diets containing pea protein. There is a noticeable difference in the proportion of microbial amino acids, but it should be noted that the weaner pigs were cannulated, which may have triggered a proliferation of gut microbes.

Importantly, static amino acid digestibility coefficients are not the sole determinant of the extent to which amino acids are absorbed. The intestinal uptake of an amino acid is a function of its dietary concentration, the dietary feed intake and the digestibility coefficient. Moreover, the likelihood is that high voluntary feed intakes will depress digestibility coefficients by accelerating gut passage rates and the reverse applies for low voluntary feed intakes. In a recent study [86], ileal digestibility coefficients of ten amino acids were negatively correlated with feed intakes to significant extents as was the case (*r* = −0.428; *p* < 0.04) with the total of sixteen amino acids.

### 3.2. Impacts of Reducing Dietary Crude Protein on Amino Acid Digestibilities

The effects of reducing dietary crude protein on amino acid digestibilities are of obvious relevance. The multiple linear regression equation (*r* = 0.837; *p* = 0.003) derived from Chrystal et al. [38] data for concentrations of amino acid remaining in distal ileal digesta with molar amino acid proportions of microbial protein, amylase and mucin (Table 6) in birds offered 222 g/kg CP, maize-based diets is as follows:y = 0.296 + 0.318 × microbial + 0.041 × amylase + 0.008 × mucin.(1)

This contrasts with the regression (*r* = 0.834; *p* = 0.003) in birds offered 165 g/kg CP diets where:y = 0.009 + 0.047 × microbial + 0.319 × amylase + 0.008 × mucin.(2)

Taken together, the two relationships indicate that reductions in dietary CP decrease the gut microbial population and increase pancreatic secretions of amylase but have very little impact on mucin secretion. Pursuant to the dietary CP reductions, the increase in amylase secretions is consistent with the 37.4% (448 versus 326 g/kg) increase in analysed starch concentrations. The 24.6% (169 versus 224 g/kg) decrease in analysed protein concentrations may have depressed proliferation of gut microflora. While these factors are in play, the overall impacts of reduced-CP diets on amino acid digestibilities are both conflicting and complex.

In another study, stepwise reductions in dietary CP in maize-based diets from 210 to 165 g/kg CP linearly increased (*r* = −0.556; *p* = 0.002) mean distal jejunal digestibility coefficients of 17 amino acids by 29.4% (0.594 versus 0.459) at 35 days post-hatch [87]. In the distal ileum, the corresponding increase of 6.18% (0.790 versus 0.744) was also significant and the linear effect approached significance (*r* = −0.355; *p* = 0.064). The linear effect was significant for six individual amino acids, arginine, isoleucine, threonine, valine, aspartic acid and proline, where the first four were present in the 165 g/kg CP diet as both non-bound and protein-bound entities. Thus, the [87] study suggests that reducing dietary CP has a positive impact in amino acid digestibilities. This could partially be attributed to the increase from 4.50 to 21.52 g/kg in dietary concentrations of non-bound amino acids that accompanied the decrease in dietary CP from 210 to 165 g/kg, given that non-bound amino acids are notionally 100% digestible [39].

However, contrasting outcomes were reported in Chrystal et al. [38]. Reductions in CP from 222 to 165 g/kg in maize-based diets depressed mean distal ileal digestibility coefficients of 15 amino acids by 3.26% (0.771 versus 0.797). However, digestibility coefficients of six amino acids were increased to significant extents based on pairwise comparisons, seven amino acids had their digestibility coefficients significantly decreased, whereas the digestibility of glutamic acid (*p* = 0.417) and proline (*p* = 0.197) was not altered. Moreover, the increase in dietary concentrations of non-bound amino acids, from 7.23 to 38.49 g/kg, was not associated with the profound differences in amino acid digestibilities observed. For example, threonine digestibility increased by 15.4% (0.818 versus 0.709; *p* < 0.001) following the reduction in dietary CP and an additional 2.92 g/kg non-bound threonine was included in the 165 g/kg CP diet. However, glycine digestibility was decreased by 11.4% (0.661 versus 0.746; *p* < 0.001) despite an additional inclusion of 3.25 g/kg non-bound glycine in the 165 g/kg CP diet.

This is not to suggest that the [38] outcomes are typical, although the percentage differences in amino acid digestibilities in wheat-based diets recorded in the same study were very similar to maize given the highly significant linear relationship (*r* = 0.925; *p* < 0.0001). However, disparities in amino acid digestibilities pursuant to dietary CP reductions certainly complicate the quest of developing accurate ‘ideal protein ratios’ to meet amino acid requirements in birds offered reduced-CP diets.

It is worth noting that reduced-CP diets have been shown to influence ileal amino acid digestibility and gene expression of digestive enzymes in pigs [88]. In finisher pigs, a reduction in dietary CP from 160 to 100 g/kg increased average apparent ileal digestibility coefficients of 18 amino acids by 6.34% (0.804 versus 0.757), where non-significant responses were confined to aspartic acid, histidine, leucine and phenylalanine. Modest inclusions of non-bound amino acids in the 100 g/kg CP diet were limited to lysine, methionine, threonine and tryptophan and, collectively, their digestibility was increased by 6.83%. The transition to the 100 g/kg CP diet depressed daily feed intake by 9.61% (2548 versus 2819 g/day). This may have retarded gut passage rates and contributed to enhanced amino acid digestibilities.

### 3.3. Impacts of Exogenous Enzymes on Amino Acid Digestibilities

Presently, addition of exogenous enzymes to broiler diets is a totally routine practice and the capacity of phytate-degrading enzymes to enhance amino acid digestibilities is established. The inclusion of 1000 FTU/kg phytase increased average ileal digestibility coefficients of 17 amino acids by 12.3% (0.840 versus 0.748; *p* < 0.001) in maize-based diets [89]. Increases in digestibility of essential amino acids ranged from 5.45% (0.928 versus 0.880) for methionine to 15.7% (0.765 versus 0.661) for threonine, which is a highly typical pattern. Subsequently, Truong et al. [90] reported that 500 FTU/kg phytase in maize-based diets increased distal ileal digestibility coefficients of 16 amino acids by 7.24% (0.904 versus 0.843). However, the corresponding increase in the proximal jejunum was 49.5% (0.719 versus 0.481) with intermediate increases of 20.4% and 9.07% in the distal jejunum and proximal ileum, respectively. Thus, the [90] study showed that in addition to phytase increasing the extent of amino acid digestion, it was also generating a proximal ‘shift’ in the sites of amino acid absorption along the small intestine, thereby influencing starch and protein digestive dynamics.

There are mechanisms whereby phytases enhance amino acid digestibilities probably have two phases [91]. Firstly, phytases promote protein digestion in the gut lumen essentially by preventing the de novo formation of binary and ternary protein-phytate complexes at more than, or less than, the isoelectric point (iP) of protein, respectively. Thus, both the pH along the gut and the iP of protein is critical. The iP of protein is in the order of 5.5 in broad terms, however, the iP of protein meals are lower and the iP of feed grain proteins are higher than an iP of 5.5. For example, the iP of soy protein is 4.70 as opposed to the iP of maize protein of 6.20 [92]. The pH of the proventriculus (1.98) and gizzard (3.14) are lower than protein iP [58] and it is possible that the amplitude of this differential will dictate the intensity of binary protein-phytate complex formation. If so, the acid binding capacity of feedstuffs and diets [93] assume importance.

Secondly, the likelihood is that phytases promote intestinal uptakes of amino acids via Na^+^-dependent transporters by increasing the activity of the ‘sodium pump’ in the basolateral membrane of enterocytes. The Na-sparing effect of exogenous phytases has been established [90,94,95], which may result in higher Na cytoplasmic concentrations within enterocytes that are prerequisites for sodium pump activity [96]. Pivotally, exogenous phytase has been shown by [97] to increase Na^+^,K^+^-ATPase or sodium pump activity in the duodenum and jejunum of broiler chickens. The addition of 1000 FTU/kg phytase to maize-based diets containing 2.2 g/kg phytate-P significantly increased sodium pump concentrations in the duodenum by 17.6% (10.01 versus 8.51 µmol/mg) and jejunal activity by 18.4% (13.59 versus 11.48 µmol/mg) in broiler chickens.

Exogenous proteases and non-starch polysaccharide-degrading enzymes (NSP) may be used in conjunction with phytases. The addition of a protease to a sorghum-based diet was shown to increase mean apparent ileal digestibility coefficients of 16 amino acids by 4.71% (0.822 versus 0.785; *p* = 0.025) and significantly increased digestibility coefficients of 13 individual amino acids with the exception of arginine, lysine and methionine [98]. In addition, protease increased mean digestion rate constants of 16 amino acids by 29.9% (3.47 versus 2.67 × 10^−2^min^−1^). Combined inclusions of phytate- and NSP-degrading feed enzymes have been shown to generate synergistic responses in wheat-based diets [99].

## 4. Amino Acid Transition across the Gut Mucosa

Pursuant to their intestinal uptakes, the post-enteral availability of amino acids is profoundly influenced by their transition across enterocytes of the small intestinal mucosa. The post-enteral availability of amino acids is substantially reduced by their entry into either anabolic or catabolic pathways in the gut mucosa, rather than entering the portal circulation [100]. This ‘diversion’ is illustrated by a 32.6% overall decline in the net portal flux of amino acids in pigs. These ‘losses’ ranged from 8.0% in arginine to 54.5% in threonine, however, the net portal flux of glutamic and aspartic acids was negligible, whereas alanine was synthesised in the small intestine [101]. Amino acids may be synthesised into proteins to maintain gut integrity or serve as precursors for digestive enzymes, mucin, nucleotides, polyamines and amino acids. Alternatively, amino acids are subject to catabolism in the gut mucosa to meet the profuse energy needs of the gut [102]. The quantitative extent of amino acid catabolism in poultry is not known, but 18% of amino acids are catabolised in weaner pigs based on the net portal outflow of ammonia relative to amino acid intakes and amino acid catabolism exceeded the incorporation of amino acids into mucosal protein [103]. Reeds et al. [84] concluded that amino acids are critical energy sources for the intestinal mucosa and questioned if this was subject to manipulation by nutritional strategies.

Glutamine and ketone bodies were identified as major post-absorptive energy substrates in the small intestine of rats by [104], while glutamine, glucose and, to a lesser extent, ketone bodies are leading respiratory fuels in human enterocytes [105]. Glucose and glutamine were found to provide similar proportions of energy to the gut mucosa in rats and it appeared that energy was derived more efficiently from glucose [106]. In poultry, glucose, glutamate and glutamine were reported to be catabolised in avian enterocytes [107]. The metabolism of glutamate, glutamine, aspartate, asparagine, glucose and ketone bodies in the gut mucosa of chickens have been investigated, whereby glucose stimulated respiration and was converted mainly into lactate, but this glycolysis was reduced in the presence of glutamine and aspartate [108].

The extent to which either amino acids or glucose (and alternative energy substrates) are catabolised in the avian gut mucosa is potentially important but is essentially an unknown. Given glucose is an important energy substrate, it may be possible to manipulate the “catabolic ratio” of glucose to amino acids. If more glucose undergoes catabolism, then more amino acids would be spared from catabolism to enter the portal circulation and energy would be derived more efficiently from glucose [106]. Some backing for this proposition was provided by Enting et al. [109], where a slowly digestible starch (pea) generated superior FCR to a rapidly digestible starch (tapioca) by up to 4.33% (1.414 versus 1.478) in diets with a low digestible lysine level from 9 to 18 days post-hatch. It was concluded that a slowly digestible starch seems to have an amino acid sparing effect.

Of relevance is that the transition from a rapidly (maize) to slowly (pea) digestible starch source significantly increased the net portal flux of lysine by 28.6% (31.0 versus 24.1 mmol) and tyrosine by 23.7% (16.7 versus 13.5 mmol) and tended to increase (*p* < 0.10) the net portal flux of another nine amino acids in pigs [110]. It was concluded that more glucose was used as an oxidative substrate by the gut mucosa following the feeding of slowly digestible pea starch, thereby permitting more amino acids to enter the portal circulation as they were spared from catabolism in the gut mucosa. Somewhat contradictory findings in pigs have been reported where rapid sticky rice starch supported 7.21% (0.818 versus 0.763) higher mean apparent ileal amino acid digestibility coefficients and 14.2% (0.258 versus 0.226 mmol/L) higher mean amino acid concentrations in systemic plasma than slow maize starch [111]. Thus, the question posed by Reeds et al. [100] as to the possible manipulation of the transit of amino acids across the gut mucosa remains pertinent and the potential impact of starch is evident.

## 5. Bioequivalence of Non-Bound and Protein-Bound Amino Acids

Non-bound and protein-bound amino acids are not likely to be fully bioequivalent in poultry, if only because intestinal uptakes of non-bound amino acids are more rapid that their protein-bound counterparts as they do not require prior digestion. This was demonstrated in several pig studies by Ted Batterham and his colleagues [112,113,114,115]. The utilisation of non-bound lysine HCl was compromised by feeding pigs on a restricted basis rather than ad libitum and this was attributed to the asynchronous availability of lysine HCl and protein-bound amino acids at sites of protein synthesis in restricted fed pigs. Net post-prandial lysine and threonine concentrations in the portal circulation of pigs fed on a once daily basis were determined by Yen et al. [116]. One hour post-prandial portal concentrations of lysine (120 versus 60 mg/L) and threonine (75 versus 45 mg/L) were approximately double in pigs offered diets containing both amino acids as non-bound entities in comparison to diets without non-bound amino acids. It was concluded that non-bound amino acids are absorbed more rapidly but there is the inference that the rapid absorption may have spared lysine and threonine from catabolism in the gut mucosa.

Comparable data has not been generated in poultry and it would be extremely difficult, but highly desirable, to determine the net portal flux of amino acids in broiler chickens. Free plasma concentrations of amino acids in the portal (anterior mesenteric vein) circulation of broiler chickens were determined in [117]. Effectively, this is the gross portal flux where free amino acid concentrations were 14.7% higher (930 versus 811 μg/mL than in the systemic (brachial vein) circulation. However, their relative concentrations were remarkably similar, which is indicative of the extent that the gut mucosa derives amino acids from the arterial circulation in addition to the gut lumen [118].

Support for our contention that non-bound and protein-bound amino acids are not bioequivalent may be garnered from Chrystal et al. [119]. In this study the CP of maize-based diets was reduced from 208 to 193, 179 to 165 g/kg with corresponding increases in non-bound amino acid inclusions from 5.30 to 10.04, 15.99 to 21.96 g/kg, which were offered to birds from 14 to 35 days post-hatch. These dietary modifications did not linearly influence (*r* = 0.073 *p* > 0.65) the mean apparent ileal digestibility coefficients of 16 amino acids at 35 days post-hatch and methionine was the only individual amino acid impacted with a 4.73% (0.841 versus 0.803) increase in ileal digestibility. Concentrations of 20 free amino acids in systemic plasma were determined at 34 days post-hatch. Plasma concentrations of eight amino acids (Ile, Trp, Ala, Asn, Gln, Glu, Gly, Pro) were not influenced by dietary CP reductions. The dietary CP reductions linearly decreased plasma concentrations of eight amino acids (Arg, His, Leu, Phe, Asp, Cys, Ser, Tyr), by up to 23.3% (23.7 versus 30.9 μg/mL) collectively. However, dietary CP reductions linearly increased plasma concentrations of lysine by up to 45.8% (80.5 versus 55.2 μg/mL), methionine by 24.0% (15.5 versus 12.5 μg/mL), threonine by 44.1% (79.1 versus 54.9 μg/mL) and valine by 18.8% (30.9 versus 26.0 μg/mL). Collectively, there was a 38.4% (51.5 versus 37.2 μg/mL) increase in concentrations of these four critical amino acids. Importantly, in the 165 g/kg CP diet, non-bound lysine represented 49.0% of the total analysed dietary concentration, non-bound methionine 70.5%, non-bound threonine 37.6% and non-bound valine 32.4%. Lysine, methionine, threonine and valine are foremost limiting amino acids for broiler growth performance. Their increased concentrations in systemic plasma are curious given that weight gains of birds were not linearly influenced (*r* = 0.307; *p* = 0.099) by graded reductions in dietary CP from 208 to 165 g/kg. This suggests that their increased plasma concentrations did not stem from decreased net protein synthesis, but the genesis of their increased plasma concentrations stemmed from increased transition across the gut mucosa of non-bound amino acids.

Instructively, the postprandial oxidative losses of egg white protein, an equivalent mix of non-bound amino acids and two intermediate blends were determined using [^13^CO_2_] breath tests by Nolles et al. [120]. In rats, postprandial oxidative losses of non-bound leucine were significantly higher than protein-derived leucine on the fifth day. The researchers concluded that non-bound and protein-bound amino acids were metabolised independently. It is highly likely that the post-enteral availability of non-bound versus protein-bound amino acids also differ in broiler chickens. The relatively rapid absorption of non-bound amino acids probably has several implications, including an amino acid sparing effect in the gut mucosa in anterior segments of the small intestine. Therefore, that non-bound and protein-bound amino acids are not fully bioequivalent represents a tangible challenge to the successful development of reduced-CP diets because of their high non-bound amino acid inclusion levels.

## 6. Functional Amino Acids

Amino acids are ‘building-blocks’ of protein, however, amino acids additionally perform a multiplicity of roles as reviewed by Wu [13,121] and his colleagues [27]. Functional amino acids regulate key pathways, including cell signalling involving protein kinases, modulating gene expression and promoting protein synthesis in the small intestine and skeletal muscle. Functional amino acids are substrates for the synthesis of physiologically important, low molecular weight substances, including glutathione, carnitine serotonin and thyroid hormones.

These additional functions are reflected in the differences in the amino acid profile of whole-body protein of broiler chickens when compared to the recommended amino acid profile of their diets [27]. There are, for example, higher relative concentrations of so-called essential (methionine, cysteine, threonine, isoleucine, valine) and non-essential (glutamic acid, glutamine, proline) amino acids in typical diets than are present in whole-body protein. The functionality of amino acids is blurring the distinction between “essential” and “non-essential” because the rate of biosynthesis of some non-essential amino acids may not be adequate. This could be the case for the synthesis of glycine from threonine in poultry. Theoretically, threonine dehydrogenase can convert threonine to glycine and acetyl CoA [122] and this pathway is probably dominant in avian species [123]. However, this enzymatic conversion may not be sufficiently rapid in practice [124]. Threonine is an abundant amino acid in endogenous secretions and especially avian mucin [75]. Curiously, elevated free threonine concentrations in systemic plasma have consistently been observed in birds offered reduced-CP diets, which is an issue addressed in Macelline et al. [125]. It appears that free plasma threonine levels may be a valuable indicator of the precision with which reduced-CP diets are formulated and the extent to which they are accommodated by broiler chickens.

This section is limited to a brief discussion of phenylalanine and tyrosine to provide examples of their functionality. Phenylalanine and tyrosine (and tryptophan) are classified as aromatic amino acids and various recommendations have been made for phenylalanine plus tyrosine requirements in broiler diets. These include a relativity to lysine of 112 for young birds [126] and 60 phenylalanine and 45 tyrosine for 22 to 42 day-old birds; that is a total of 105 of which 57% is met by phenylalanine [27]. Phenylalanine may be converted to tyrosine by hepatic and renal phenylalanine hydroxylase activity in poultry [127]. As a functional amino acid, phenylalanine is required for the synthesis of thyroid hormones such as triiodothyronine and tetraiodothyronine, which are involved in numerous metabolic processes, including electrolyte transportation and thermogenesis in poultry [128]. Phenylalanine and tyrosine are also involved in the synthesis of catecholamines, including dopamine, in cerebral tissue [129]. Usually, feather pecking is a problem confined to laying hens, however, increased incidences of feather pecking in broiler chickens have been observed following reductions in dietary CP [84]. It was suggested that feather pecking may have been triggered by inadequate aromatic amino acid plasma levels of phenylalanine and tyrosine, precursors of dopamine, and tryptophan, a precursor of serotonin, resulting in deficient levels of these ‘feel-good’ neurotransmitters [130], which have been associated with aggressive behaviour in poultry [131]. It is then relevant that the reduction in dietary CP from 215 to 162.5 g/kg CP prompted declines in analysed concentrations of 10.8 to 6.9 g/kg in phenylalanine and 5.3 to 3.2 g/kg in tyrosine in the Greenhalgh et al. [84] study, which may have caused the feather pecking observed. Probably more attention should be paid to phenylalanine and tyrosine concentrations in reduced-CP diets than has been the case.

## 7. Protein Accretion

Protein accretion is the net result of protein turnover and this topic and its determination in animals has been reviewed over several decades [132,133,134]. The small difference is the balance between protein deposition and protein degradation represents protein accretion, or net protein synthesis and growth. Protein turnover is a continuous, dynamic process taking place mainly in skeletal muscles and the rapid accumulation of breast-muscle protein in rapidly growing chicks is largely due to a marked decrease in the fractional rate of degradation [135]. This was subsequently confirmed by Tesseraud et al. [136] in a comparison of fast- and slow-growing broiler chickens. While slow-growing chickens had a higher fractional rate of protein synthesis (28 versus 22%/day); the fractional protein degradation rate of slow-growing chickens clearly exceeded (16.6 versus 10.3%/day) their fast-growing counterparts. Thus, increased protein accretion stems more from a reduction in protein degradation than a promotion in protein deposition. Protein degradation appears highly counterproductive, but fundamentally muscle protein is degraded to maintain the quality of intracellular proteins by eliminating misfolded or damaged polypeptides [137]. Nevertheless, protein synthesis is an energy demanding process as an input of 5.35 kJ is required for each gram of protein synthesised in broiler chickens [138] and decreased protein degradation is more energetically efficient than increased protein deposition [139].

The metabolism of surplus amino acids arising from protein degradation or dietary excesses was considered by Bender [140], and Sklan and Noy [141] investigated the anabolism and catabolism of amino acids in growing chicks. Deposition of carcass protein was governed by the limiting amino acid at any one time and surplus amino acids to requirements for protein accretion underwent catabolism, which is accompanied by energy costs. As discussed by Selle et al. [142], costs of deamination are invited by amino acid catabolism, partially because ammonia (NH_3_) is generated and has to be detoxified [143]. Ultimately, NH_3_-N is excreted via the kidneys as uric acid-N but uric acid synthesis via a Krebs cycle demands glycine and energy inputs [144]. The possibility is that insufficient NH_3_ detoxification impedes the growth performance of broilers offered reduced-CP diets, likely because of inadequate dietary glycine and serine levels to drive the Krebs uric acid cycle to eliminate NH_3_-N as uric acid-N [145].

Interestingly, protein degradation rates strongly correlated with feed conversion efficiency have been reported in four selected lines of broiler chickens [146]. The positive linear regressions were significant at both 14 days (*r* = 0.72; *p* < 0.001) and 42 days (*r* = 0.51; *p* < 0.001) post-hatch. Further, Urdaneta-Rincon and Leeson [147] examined protein turnover in skeletal muscle (*Pectoralis major*) in 21-day-old broiler chickens offered fourteen diets with graded levels of lysine and crude protein. These researchers determined absolute synthesis and breakdown rates of protein and therefore the differences represent net protein synthesis accretion. Absolute protein synthesis averaged 625.8 mg/day, protein breakdown averaged 321.4 mg/day and net protein synthesis averaged 304.4 mg/day, which ranged from 192 to 423 mg/day. Importantly, it may be deduced from the published treatment means data that increasing net protein synthesis rates were quadratically (*r* = 0.798; *p* < 0.001) associated with FCR improvements as shown in Figure 1. The suggestion was put forward in [146] that there is probably a positive genetic association between efficiencies of food utilisation and protein metabolism, which is subsequently supported by the outcomes in [147].

The incidence of breast muscle myopathies, including white striping, wooden breast and spaghetti meat, in broiler chickens, has been an increasing problem for the chicken-meat industry over the past decade [148]. White striping is the appearance of white striations paralleling muscle fibres in breast (and thigh) muscle and is associated with an increase in fat but a decrease in protein. The problem of white striping was investigated by Vignale et al. [149], which linked white striping myopathies to increased fractional protein breakdown rates in breast muscle. Clearly, in both specific and general terms, any strategies that can be developed to diminish rates of protein degradation in skeletal muscle would be highly advantageous for chicken-meat production.

The importance of protein in chicken-meat production is reflected in the exponential models developed by Eits et al. [150,151] to predict growth rate, feed conversion and carcass and breast meat yield of broiler chickens as a function of dietary balanced protein content. Dietary balanced protein was defined as the dietary concentration of digestible lysine (g/kg) in the context of nutritionally adequate diets. These models provide nutritionists with the choice of formulating diets to generate either maximum broiler performance or maximum profit, where electing to sacrifice some performance can increase the profitability of an enterprise.

## 8. Starch-Protein Digestive Dynamics

Instructively, Moughan [152] concluded that the efficiency of protein synthesis is markedly influenced by the harmonisation of the provision of amino acids and non-amino energy supplying compounds to sites of protein synthesis. Glucose, whether derived extrinsically from dietary starch or intrinsically from gluconeogenesis, is the dominant, but not the only, energy source for broiler chickens. Therefore, starch–protein digestive dynamics have been reviewed [153,154] and a third review by Liu and Selle [155] emphasised inclusions of non-bound amino acids in reduced-CP diets within this context. Protein deposition cannot be considered in isolation from the starch and glucose contributions into this process. Body reserves of free amino acids are small, which means the timing of inputs of amino acids and carbohydrates can markedly impact protein synthesis and feed conversion efficiency [156]. It is relevant that the circulating pool size of amino acids has been shown to be diminished from reductions in dietary CP levels in diets offered to pigs [157]. This was attributed to decreased expression of specific transporters in the jejunum and decreased absorption of amino acids, despite increased expression for PepT−1 and, presumably, increased intestinal uptakes of oligopeptides.

Digestibility coefficients are static measurements and cannot account for different absorption rates of nutrients, including glucose and amino acids [158]. Moreover, broiler chickens are intermittent rather than continuous feeders [158]. The extent to which broiler chickens held under practical conditions with ‘lights-off’ periods without illumination can accommodate different rates of amino acid absorption and post-enteral amino acid availability is a real question. However, it is possible to deduce apparent disappearance rates (g/bird/day) from small intestinal segments of the critical macronutrients, starch and protein, from daily feed intakes, dietary concentrations and digestibility coefficients. The potential confounding effect of voluntary feed intakes on growth performance parameters can be eliminated by calculating starch:protein disappearance rate ratios. As an example, broiler chicks were offered six diets based on a red sorghum (600 g/kg) from 7 to 28 days post-hatch [159]. The six dietary treatments were offered as mash, intact pellets steam-conditioned at 65, 80 and 95 °C, reground mash following pelleting at 95 °C, and the 80 °C diet with an exogenous protease added. Individual distal ileal starch and protein disappearance rates calculated retrospectively were not linearly or quadratically related to FCR. However, distal ileal starch:protein disappearance rate ratios were quadratically related (*r* = 0.458; *p* = 0.0102) to FCR as expressed in the following equation:y_(FCR)_ = 5.239 + 0.980 × ratio^2^ − 3.823 × ratio(3)

It may be deduced from the equation that a starch:protein disappearance rate ratio of 1.95 generated the minimum FCR of 1.511, as shown in Figure 2. Similar outcomes were reported [160] where proximal jejunal starch:protein disappearance rate ratios were quadratically related to weight gain (*r* = 0.849; *p* < 0.001) and FCR (*r* = 0.838; *p* < 0.001) in broiler chickens at 28 days post-hatch. These quadratic relationships indicate that there is an ideal balance between intestinal uptakes of protein as oligopeptides and monomeric amino acids and energy as glucose from starch to generate optimal growth performance. They further indicate that the digestive dynamics of protein and starch should not be considered in isolation.

However, the post-enteral availability of glucose and amino acids is subject to the adaptive regulation of their uptakes along the small intestine, which probably constitutes a neglected research area [161]. Arguably, this applies to broiler chickens in particular. This was evident in a reduced-CP diet experiment with very high dietary starch levels [83] where apparent proximal ileal digestibility coefficients of starch were negatively correlated with twelve amino acids, including all ‘essential’ amino acids to significant extents. Significant correlations numbered two in proximal jejunum, nine in distal jejunum, eleven in distal ileum and arginine were negatively correlated with starch in all four small intestinal segments. Moss et al. [83] advanced the rationale that glucose and amino acids were competing for co-absorption with sodium (Na) via their respective Na^+^-dependent transport systems. Glucose is principally absorbed via the Na^+^-dependent transporter, SGLT-1 [162] whereas amino acids are absorbed via fourteen Na^+^-dependent transport systems with overlapping specificities, the majority of which are involved with intestinal uptakes of neutral amino acids [50]. However, all these transport systems are driven by the activity of the ‘sodium pump’ (Na^+^,/K^+^-ATPase) in the basolateral membrane of enterocytes [163].

The digestive dynamics of starch and protein has important impacts on the transition of amino acids and glucose across the gut mucosa and their post-enteral availability at sites of protein synthesis. The provision of some slowly digestible starch in broiler diets has been shown to advantage feed efficiency and breast meat yield [164]. This could be partially attributable to slowly digestible starch sparing amino acids from catabolism in the gut mucosa [109]. Reciprocally, a “rapid-protein” diet fortified with casein and additional non-bound amino acids (arginine, isoleucine, lysine, methionine, threonine, tryptophan) was compared with a “slow-protein” diet based on soybean meal, canola meal, maize and limited quantities of non-bound amino acids (lysine, methionine, threonine) in Truong et al. [117]. The “rapid-protein” diet numerically enhanced weight gain and FCR and significantly increased free plasma concentrations of methionine, threonine and proline in the portal circulation. Six amino acids were present in the summit diet in both non-bound and protein-bound entities. The transition from slow to rapid protein diets increased their aggregate concentrations in the portal circulation by 17.4% (257 versus 219 μg/mL). In contrast, portal concentrations of twelve protein-bound amino acids in the rapid protein diet fractionally decreased from 715 to 709 μg/mL. While not conclusive, this contrast is consistent with the possibility that non-bound amino acids are less likely to be catabolised in the gut mucosa than their protein-bound counterparts. Non-bound amino acids may be spared from catabolism because they are rapidly absorbed in the anterior small intestine, where most starch is digested and more glucose is available as an alternative energy substrate. Conversely, slowly digestible starch would make more glucose available in the posterior small intestine to be oxidized, again potentially sparing amino acids from catabolism. The importance of understanding kinetics of digestion and its complexity was emphasised in a recent, comprehensive study reported by Pedersen et al. [165].

Feedstuffs with predetermined starch and protein digestion rate constants were incorporated into six diets that were offered to broilers from 7 to 35 days post-hatch [166]. Retarding starch digestion rates quadratically (*r* = 0.634; *p* < 0.005) improved feed conversion efficiency, as shown in Figure 3. Alternatively, accelerating protein digestion rates quadratically (*r* = 0.656; *p* < 0.001) improved FCR. Moreover, condensing the starch:protein digestion rate ratio quadratically (*r* = 0.648; *p* < 0.001) improved FCR where:y_(FCR)_ = 1.712 + 0.095 *× ratio^2^ − 0.316 × ratio(4)

Therefore, a starch:protein digestion rate ratio of 1.663 would support the minimum FCR of 1.450 from 7 to 35 days post-hatch. These outcomes support the concepts that the provision of both slowly digestible starch and rapidly digestible protein in broiler diets is advantageous. They further support that starch and protein digestive dynamics should be considered in tandem.

## 9. Conclusions

The sustainable production of broiler chickens in the future will stem largely from improvements in the efficiency of converting dietary proteins and amino acids into chicken-meat protein. These improvements could be generated by both selection programs and nutritional strategies. However, both approaches are thwarted by inadequacies in our knowledge of the relevant physiological and biochemical pathways in broiler chickens, which is obvious when compared to humans and other food-producing animal species. The development and adoption of reduced-CP broiler diets holds enormous promise for sustainable chicken-meat production and this opportunity has been central to this review. While the advantages that would flow from the adoption of viable reduced-CP broiler diets are irrefutable, the obstacles that stand in the way of their development are tangible. Vagaries in apparent amino acid digestibility coefficients and disappearance rates pursuant to reductions in dietary CP are in evidence, which complicate the formulation of reduced-CP diets based on ideal amino acid ratios. It is our contention that ideal amino acid ratios for a reduced-CP diet will differ from diets with standard CP concentrations. However, identification of ideal amino acid ratios for reduced-CP diets is a formidable challenge. Inclusion of feed grains increase in reduced-CP diets and the properties of starch in a given feed grain, including starch digestion rates, appear to hold importance. It has been shown that maize is a more suitable feed grain than wheat as the basis of reduced-CP diets. Somewhat ironically, the genesis of this advantage appears to be the lower protein content of maize and lower non-bound amino acid inclusions in maize-based, reduced-CP diets. A problematic issue is the bioequivalence of non-bound versus protein-bound amino acids where the likelihood is that they are not identical. Similarly, the metabolic fate of amino acids in their transition across enterocytes of the gut mucosa and their interactions with alternative energy substrates, especially glucose, is an area which requires clarification in poultry. Another issue is whether or not the conversion of essential amino acids to non-essential amino acids occurs at sufficiently rapid rates in broiler chickens to meet requirements and emphasise the importance of the many functional roles played by amino acids, over and above protein accretion. The all-important differential between protein synthesis and protein degradation, or net protein synthesis, in skeletal muscle of broiler chickens demands further investigation. Concentrations of free amino acids in portal and/or systemic plasma should provide insights into the post-enteral availability of amino acids, but interpretation of this data is not straightforward as it reflects a complex and kinetic position. The metabolism of protein and energy in broiler chickens is inextricably linked, where protein synthesis is just one example, and we need to advance our comprehension of both factors in tandem. Indeed, it is our contention that a better appreciation of starch and protein digestive dynamics in broiler chickens is a necessity if progress is to be achieved in the development of reduced-CP diets. In this respect, free threonine concentrations in systemic plasma may be indicative. Finally, more fundamental and applied research targeting areas where our knowledge is lacking needs to be completed if the objective of sustainable chicken-meat production via reduced-CP diets is to be realised.

## Figures and Tables

**Figure 1 animals-11-02288-f001:**
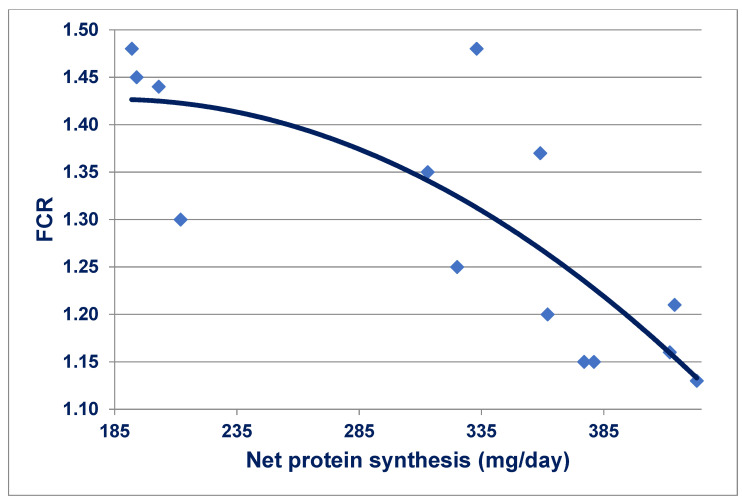
Quadratic relationship (*r* = 0.798; *p* < 0.001) between net protein synthesis in skeletal muscle (*Pectoralis major*) and FCR in 21 day-old broiler chickens [147].

**Figure 2 animals-11-02288-f002:**
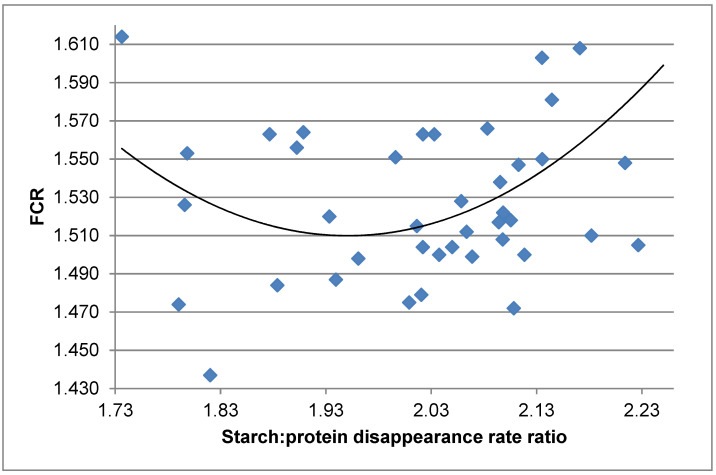
Quadratic relationship (*r* = 0.458; *p* = 0.0102) between starch:protein disappearance rate ratios in the distal ileum and FCR in broiler chickens offered sorghum-based diets from 7 to 28 days post-hatch [154,159].

**Figure 3 animals-11-02288-f003:**
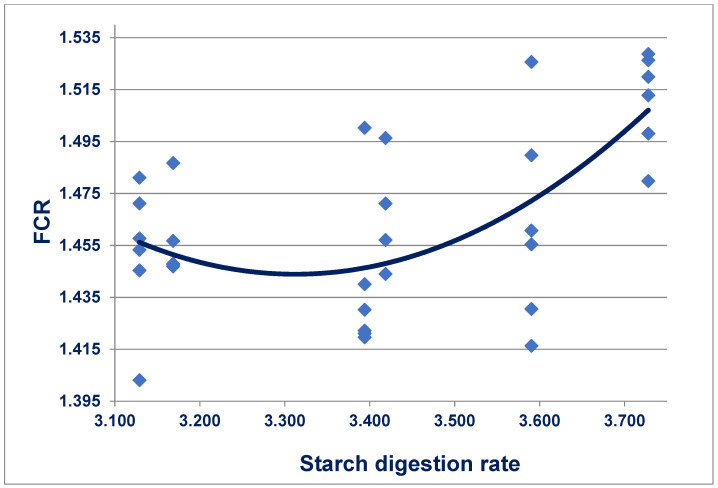
Quadratic relationship (*r* = 0.634; *p* < 0.005) between starch digestion rate constants and FCR where: y = 5.470 − 2.431 x starch + 0.367 x starch^2^ from 7 to 35 days post-hatch [165].

**Table 1 animals-11-02288-t001:** Ten examples of ideal protein ratios for broiler chickens published between 1965 and 2014.

Amino Acid	Dean and Scott [18]	ARC [19]	AEC [20]	NRC [21]	NRC [22]	Baker [23]	Roth et al. [24]	Tillman and Dozier [25]	Wecke and Liebert [26]	Wu [27]
Arginine	98	94	105	120	114	105	108	105	105	108
Histidine	27	44	39	29	32	32	38		34	35
Isoleucine	71	77	72	67	73	67	63	66	69	69
Leucine	107	134	133	113	109	109	108		110	109
Lysine	100	100	100	100	100	100	100	100	100	100
Methionine	40	44	45	42	46	36	37	45	40	42
Phenylalanine	64	77	67	60	66		62		66	60
Threonine	58	67	62	67	73	67	67	70	66	70
Tryptophan	20	19	17	19	18		14	16	16	17
Valine	73	89	79	68	72	77	81	75	80	80
Alanine										102
Aspartic acid										122
Cysteine	31	40	31	36	36	36	33	36	34	33
Glutamic acid	1071									306
Glycine	143									176
Proline	89									184
Serine										69
Tyrosine	56						59		54	45

**Table 2 animals-11-02288-t002:** Amino acid compositions of selected feedstuffs and profiles relative to lysine (100) in comparison to Texas A&M optimal ratios for broiler chickens from 21 to 42 days post-hatch.

Amino Acid	Optimal Ratio ^1^	Soybean Meal ^2^		Fishmeal ^2^		Meat and Bone Meal ^2^		Maize ^2^		Sorghum ^2^	
		g/kg	Ratio	g/kg	Ratio	g/kg	Ratio	g/kg	Ratio	g/kg	Ratio
Arginine	108	31.2	109	34.0	92	48.5	116	4.1	152	31.2	186
Histidine	35	11.5	40	27.8	29	15.1	38	2.3	92	11.5	105
Isoleucine	69	21.0	73	49.1	62	32.6	61	3.8	136	21.0	173
Leucine	109	37.0	129	88.2	99	52.4	113	12.1	452	37.0	550
Lysine	100	28.7	100	74.9	100	52.9	100	2.2	100	28.7	100
Methionine	42	6.4	22	26.4	38	20.2	35	2.0	84	6.4	91
Phenylalanine	60	24.4	85	48.7	53	27.8	59	5.1	184	24.4	232
Threonine	70	20.3	71	41.0	78	41.1	77	3.2	124	20.3	145
Tryptophan	17	6.3	22	12.4	13	7.0	12	1.0	28	6.3	45
Valine	80	22.5	78	60.3	72	38.0	71	5.0	176	22.5	227
Alanine	102	20.8	72	27.7	96	50.7	151	9.6	284	20.8	436
Asparagine	56	24.2	84	25.6	55	29.2	70	3.1	140	24.2	141
Aspartic acid	66	34.0	118	38.8	82	43.4	97	3.6	172	34.0	164
Cysteine	33	6.9	24	4.3	13	6.7	16	1.9	80	6.9	86
Glutamic acid	178	45.3	158	93.8	114	60.1	128	11.8	256	45.3	536
Glutamine	128	41.1	143	112	74	39.4	89	8.5	408	41.1	386
Glycine	176	27.2	95	18.6	124	65.8	274	3.9	160	27.2	177
Proline	184	31.8	111	108	80	42.5	185	9.6	424	31.8	436
Serine	69	23.5	82	50.8	53	28.0	66	4.6	180	23.5	209
Tyrosine	45	17.2	60	50.6	45	23.6	46	4.5	172	17.2	205

^1^ Taken from Wu [27]; ^2^ Taken from Li et al. [28].

**Table 3 animals-11-02288-t003:** Apparent ileal digestibility coefficients of amino acids in the same maize-soy diet determined by five institutions, adapted from Ravindran et al. [37].

Amino Acid	Station One	Station Two	Station Three	Station Four	Station Five	Mean
Arginine	0.920	0.917	0.917	0.896	0.909	0.912
Histidine	0.887	0.885	0.884	0.868	0.872	0.879
Isoleucine	0.871	0.869	0.870	0.848	0.854	0.862
Leucine	0.885	0.879	0.888	0.868	0.873	0.879
Lysine	0.887	0.888	0.886	0.862	0.870	0.879
Methionine	0.912	0.915	0.913	0.892	0.894	0.905
Phenylalanine	0.849	0.842	0.868	0.847	0.806	0.842
Threonine	0.820	0.815	0.798	0.786	0.781	0.800
Tryptophan	0.847	0.850				0.849
Valine	0.861	0.860	0.859	0.837	0.839	0.851
Alanine	0.877	0.876	0.874	0.858	0.854	0.868
Aspartic acid	0.853	0.844	0.839	0.824	0.824	0.837
Cysteine	0.792	0.767	0.742	0.740	0.749	0.758
Glutamic acid	0.826	0.825	0.811	0.803	0.791	0.811
Glycine	0.911	0.825	0.811	0.803	0.791	0.811
Proline	0.872	0.862	0.869	0.863	0.854	0.864
Serine	0.864	0.857	0.851	0.837	0.839	0.850
Mean	0.867	0.862	0.861	0.845	0.844	0.856

**Table 4 animals-11-02288-t004:** Composition of maize- and wheat-based, diets with standard and reduced CP concentrations adopted from Chrystal et al. [38].

Feed Ingredient (g/kg)	Maize-Based Diets		Wheat-Based Diets	
	222 g/kg CP	165 g/kg CP	222 g/kg CP	165 g/kg CP
Wheat (107 g/kg)	-	-	525	751
Maize (81 g/kg)	511	721	-	-
Canola seed (220 g/kg)	60.0	60.0	60.0	60.0
Soybean meal (483 g/kg)	334	113	300	48.0
Soy oil	35.0	-	52.0	20.0
*l*-lysine HCl	1.60	8.12	2.36	9.72
*d,l*-methionine	2.67	4.53	2.75	4.81
*l*-threonine	1.18	4.10	1.59	4.93
*l*-tryptophan	-	0.79	-	0.67
*l*-valine	1.80	3.88	0.47	4.61
*l*-arginine	-	5.77	-	6.99
*l*-isoleucine	-	3.46	0.01	4.15
*l*-leucine	-	1.41	-	5.39
*l*-histidine	-	0.81	-	1.55
Glycine	0.32	3.57	0.41	3.95
*l*-serine	0.01	3.84	0.43	4.76
Sodium chloride	3.77	0.53	2.23	-
Sodium bicarbonate	0.89	5.72	2.90	6.16
Potassium carbonate	-	6.69	-	9.49
Limestone	5.96	5.82	5.92	5.74
Dicalcium phosphate	21.2	24.4	21.6	25.1
Choline chloride	0.90	0.90	0.90	0.90
Celite	20.0	20.0	20.0	20.0
Vitamin-mineral premix	2.00	2.00	2.00	2.00
Total non-bound amino acids ^1^	7.23 *(3.4%)*	38.5 *(25.4%)*	7.50 *(3.6%)*	49.4 *(32.3%)*
Soy protein ^1^	161.3 *(76.8%)*	54.6 *(36.0%)*	144.9 *(69.5%)*	23.2 *(15.2%)*
Feed grain protein ^1^	41.4 *(19.7%)*	58.4 *(38.5%)*	56.2 *(26.9%)*	80.4 *(52.5%)*

^1^ Quantities of protein source with percentage share of total shown in italics within parentheses.

**Table 5 animals-11-02288-t005:** Retention time (minutes), pH and enzyme activity (U/mL) of digesta in individual segments of gastrointestinal tract of broiler chickens. Total retention time is 338.0 min.

Item	Crop	Proventriculus	Gizzard	Duodenum	Jejunum	Ileum	Large Intestine
Retention time ^1^	7.4	4.2	50.2	7.2	59.5	86.4	123.1
pH ^2^	4.89	1.98	3.14	5.53	6.06	6.62	6.48
Trypsin ^3^	-	-	-	13.8	50.2	19.9	-
Chymotrypsin ^3^	-	-	-	7.48	13.7	4.58	-
Amylase ^3^	-	-	-	55.9	430.5	207.5	-
Lipase ^3^	-	-	-	1.4	0.41	0.05	-

^1^ Shires et al. [59]; ^2^ Shafey et al. [58]; ^3^ Ren et al. [60].

**Table 6 animals-11-02288-t006:** Amino acid sequences (molar percentage) of endogenous secretions into the digestive tract, gut microbiota and whole-body protein (WBP).

Amino Acid	Mucin ^1^	Pepsin ^2^	Trypsin ^3^	Amylase ^4^	Microbial ^5^	WPB ^6^
Arginine	2.5	1.3	1.7	6.3	3.3	4.9
Histidine	1.7	1.1	1.7	3.0	1.8	1.7
Isoleucine	3.3	6.4	6.9	4.4	4.8	3.4
Leucine	4.8	6.5	7.4	4.8	7.0	6.6
Lysine	3.7	2.5	4.3	4.4	5.4	5.2
Methionine	0.5	2.8	1.0	2.4	1.6	1.6
Phenylalanine	2.3	6.0	1.8	4.8	3.8	2.6
Threonine	19.0	7.8	4.5	5.3	6.4	3.8
Tryptophan	-	1.7	1.8	3.5	-	0.7
Valine	6.7	6.9	7.2	7.1	7.5	4.4
Alanine	5.0	4.9	6.8	6.3	11.0	9.3
Aspartic acid	7.2	11.9	10.6	14.2	11.9	7.4
Cysteine	1.3	2.3	5.2	-	1.7	1.5
Glutamic acid	9.1	7.4	7.7	8.0	13.1	11.3
Glycine	6.6	8.6	11.3	10.1	9.0	19.1
Proline	8.9	4.5	4.0	4.4	5.1	9.2
Serine	15.9	11.3	12.1	6.3	6.3	5.3
Tyrosine	2.3	6.4	3.7	4.7	2.5	1.8

^1^ Fang et al. [75]; ^2^ Bohak [76]; ^3^ Hermodson et al. [77]; ^4^ Buonocore et al. [78]; ^5^ Parsons et al. [79], Miner-Williams et al. [80]; ^6^ Wu [19].

**Table 7 animals-11-02288-t007:** Pearson correlations between amino acid sequences of mucin, pepsin, trypsin, amylase, microbial and whole-body protein (WBP).

	Mucin	Pepsin	Trypsin	Amylase	Microbial	WBP
Mucin	1.000					
Pepsin	*r* = 0.663	1.000				
	*p* = 0.006					
Trypsin	*r* = 0.494	*r* = 0.828	1.000			
	*p* = 0.044	*p* < 0.001				
Amylase	*r* = 0.265	*r* = 0.709	*r* = 0.722	1.000		
	*p* = 0.320	*p* = 0.001	*p* = 0.001			
Microbial	*r* = 0.443	*r* = 0.638	*r* = 0.701	*r* = 0.762	1.000	
	*p* = 0.075	*p* = 0.006	*p* = 0.002	*p* = 0.001		
WBP	*r* = 0.270	*r* = 0.436	*r* = 0.635	*r* = 0.608	*r* = 0.701	1.000
	*p* = 0.295	*p* = 0.070	*p* = 0.005	*p* = 0.010	*p* = 0.002	

**Table 8 animals-11-02288-t008:** Sources of amino acids in distal ileal digesta from three studies calculated by the Duvaux et al. [81] procedure.

Dietary Crude Protein (g/kg)	Sources of Amino Acids (%) in Ileal Digesta			Reference
	Dietary ^1^	Endogenous ^2^	Microbial ^3^	
213	48.3	28.9	22.8	Moss et al. [83]
222	56.4	24.2	19.3	Greenhalgh et. [84]
206	46	29.6	24.4	Greenhalgh et. [84]
187	46.1	30.1	23.8	Greenhalgh et. [84]
164	37.7	34.8	27.5	Greenhalgh et. [84]
204	60.9	22	17.1	Greenhalgh et. [84]
173	56.1	24.7	19.2	Greenhalgh et. [84]
174	45.8	30.4	23.8	Greenhalgh et. [84]
213	63	20.8	16.2	Chrystal et al. [38]
163	66.5	18.8	14.7	Chrystal et al. [38]
Mean	52.7	26.4	20.9	

^1^ Amino acid profile for dietary protein from tabulated references; ^2^ Amino acid profile for endogenous protein from Ravindran and Hendriks [85]; ^3^ Amino acid profile for microbial protein from Parsons et al. [79], Miner-Williams et al. [80].

## Data Availability

Not applicable.

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
