# Peer review of "The Dynamic Conversion of Dietary Protein and Amino Acids into Chicken-Meat Protein"

_animals, 2021, doi:10.3390/ani11082288_

Round 1

Reviewer 1 Report

The author has summarized a large number of literatures on broiler’s protein and amino acid metabolism, including the sources of protein and amino acids in feed, the digestion and absorption of protein and amino acids in the intestine, the first pass metabolism of amino acids, the intestinal mucosal amino acid metabolism, bioequivalence of non-bound and protein-bound amino acids, the role of functional amino acids, protein synthesis in broilers’ body. Authors analyzed throughout the process of feed protein into muscle protein in broilers and pointed out which aspects can be further studied to improve the conversion efficiency of chicken meat from feed protein. However, neither the abstract nor the conclusion are consistent with the main text. The two sections need to be strengthened. The balance of starch and protein in feed of broilers is not be fully discussed in the paper. The author should pay attention to the energy metabolism of broilers and indicate the future research direction of broiler nutrition in the conclusion section, besides of the application of low-protein diets.

Author Response

July 30, 2021

Ms Vicky Liu

Animals Editorial Office

Manuscript ID: Animals-1315964

The dynamic conversion of dietary protein and amino acids into chicken-meat protein

Dear Ms Liu,

                        This letter is in response to your Email of 27th July in relation to the above manuscript. We can confirm that the author list and their corresponding affiliations are correct. Also, on behalf of my co-authors, we would like to thank your Journal for the prompt attention in reviewing our manuscript and the opportunity of having it published in Animals

In accordance with Reviewer 2, the Title of the manuscript has been changed to the above Title. Our responses to all three Reviewers follow as an Attachment. As suggested, the Abstract and Conclusions have been re-written and the revised versions of both sections are attached. Scope permits few changes to be made to the Abstract; however, the Conclusions have been re-written very extensively.  The minor corrections have been attended to as outlined in the Attachment. To that end a tracked-changed version of the manuscript accompanies this cover letter, which incorporates the changes that have been made.   

We look forward to your adjudication.

Yours sincerely,

Peter H Selle BVSc PhD MRCVS

Poultry Research Foundation,

within The University of Sydney  

Attachment

Reviewer 1

Both the Abstract (to the extent possible) and Conclusions have been re-written in accordance with this Reviewer’s suggestions and, in our view, are now more consistent with the text. However, our contention is that dietary starch:protein ratios in reduced-CP diets are critical and their consideration should be retained. The energy metabolism in broiler chickens is now given some additional consideration in Conclusions.

Reviewer 2

We have endeavoured to comply with the Animals format and apologise for any shortfalls.

We have changed the Title from:

“The dynamic conversion of dietary protein and amino acids into chicken-meat in relation to other animal species”

to  

“The dynamic conversion of dietary protein and amino acids into chicken-meat protein”,

which is more succinct and accords with Reviewer 2’s suggestion. We feel “chicken-meat protein” is more apt than “chicken”, as the latter could be taken to mean whole body protein rather than skeletal muscle protein. 

Volume of soybean meal fed to broilers in USA is now defined in the track-changed manuscript

Descriptions of “Ratio A” and “Ratio B” in Section 2 are now provided

We do not see the necessity of having “P” in italics

Spelling error in L401 corrected

L425 The reference cited is correct but we have re-written the sentence in compliance

Reviewer 3

We agree with this Reviewer to the extent that it is difficult to interpret the data stemming from free amino acid concentrations in systemic (and/or portal) plasma. Perhaps this is not so much to do with their proportion of total amino acids (which is small) but, more so, the overall complexity of amino acid metabolism and its kinetics. Text changes in Conclusions now reflect this situation. 

Abstract: This review considers the conversion of dietary protein and amino acids into chicken-meat protein and to identify strategies whereby this transition may be enhanced. Viable alternatives to soybean meal would be advantageous but the increasing availability of non-bound amino acids is providing the opportunity to develop reduced-crude protein (CP) diets, to promote the sustainability of the chicken-meat industry and is the focus of this review. Digestion of protein and intestinal uptakes of amino acids is critical to broiler growth performance; however, the transition of amino acids across enterocytes of the gut mucosa is complicated by their entry into either anabolic or catabolic pathways, which reduces their post-enteral availability. Both amino acids and glucose are catabolised in enterocytes to meet the energy needs of the gut; therefore, starch and protein digestive dynamics and the possible manipulation of this ‘catabolic ratio’ assume importance. Finally, net deposition of protein in skeletal muscle is governed by the synchronised availability of amino acids and glucose at sites of protein deposition. There is a real need for more fundamental and applied research targeting areas where our knowledge is lacking relative other animal species to enhance the conversion of dietary protein and amino acids into chicken-meat protein.

Conclusions

The sustainable production of broiler chickens in the future will stem largely from improvements in the efficiency of converting dietary proteins and amino acids into chicken-meat protein. These improvements could be generated by both selection programs and nutritional strategies. However, both approaches are thwarted by in-adequacies in our knowledge of the relevant physiological and bio-chemical pathways in broiler chickens, which is obvious when compared to humans and other food-producing animal species. The development and adoption of reduced-CP broiler diets holds enormous promise for sustainable chicken-meat production and this opportunity has been central to this review. While the advantages that would flow from the adoption of viable reduced-CP broiler are irrefutable, the obstacles that stand in the way of their development are tangible. Vagaries in apparent amino acid digestibility coefficients and disappearance rates pursuant to reductions in dietary CP are in evidence, which complicate the formulation of reduced-CP diets based on ideal amino acid ratios. It is our contention that ideal amino acid ratios for a reduced-CP diet will differ from diets with standard CP concentrations; however, identification of ideal amino acid ratios for reduced–CP diets is a formidable challenge. Inclusion of feed grains increase in reduced-CP diets and the properties of starch in a given feed grain, including starch digestion rates, appear to hold importance. It has been shown that maize is a more suitable feed grain than wheat as the basis of reduced-CP diets; somewhat ironically, the genesis of this advantage appears to be the lower protein content of maize and lower non-bound amino acid inclusions in maize-based, reduced-CP diets. A problematic issue is the bioequivalence of non-bound versus protein-bound amino acids where the likelihood is that they are not identical. Similarly, the metabolic fate of amino acids in their transition across enterocytes of the gut mucosa and their interactions with alternative energy substrates, especially glucose, is an area which requires clarification in poultry. Another issue is whether or not the conversion of essential amino acids to non-essential amino acids occurs at sufficiently rapid rates in broiler chickens to meet requirements and emphasises the im-portance of the many functional roles played by amino acids over and above protein accretion. The all-important differential between protein synthesis and protein degradation, or net protein synthesis, in skeletal muscle of broiler chickens demands further investigation. Concentrations of free amino acids in portal and/or systemic plasma should provide insights into the post-enteral availability of amino acids, but interpretation of this data is not straightforward as it reflects a complex and kinetic position. The metabolism of protein and energy in broiler chickens is inextricably linked, where protein syn-thesis is just one example, and we need to advance our comprehension of both factors in tandem. Indeed, it is our contention that a better appreciation of starch and protein digestive dynamics in broiler chickens is a necessity if progress is to be achieved in the development of reduced-CP diets. In this respect, free threonine concentrations in systemic plasma may be indicative. Finally, more fundamental and applied research targeting areas where our knowledge is lacking needs to be completed if the objective of sustainable chicken-meat production via reduced-CP diets is to be realised.

Reviewer 2 Report

The manuscript by Macelline et al reviews the dynamic conversion of dietary protein and amino acids into chicken-meat in relation to other animal species and explores the transition of dietary protein and amino acid into carcass protein in broiler chicken and find a strategy to enhance this conversion. In this manuscript, the author reviews the research process of protein and amino acids sources, digestion and uptake of amino acids in the intestinal and other physiological knowledge about amino acids. This review provides new ideas for poultry amino acid research and low-protein feeding applications. The manuscript is mostly written, and contents will be of interest to readers of the Journal. There are, however, some issues that need to be corrected.

  1. The manuscript is not prepared according to the style and form for the Animals. The authors need to download the Style and Form from the Animals website and make sure they strictly follow all requirements.
  2. The review emphasis on the dynamic conversion of protein and amino acids into chicken, so the title would replace chicken-meat with chicken.
  3. In introduction part, the author displays the amount of soybean meal used in USA, please define the amount of soybean meal fed to broiler. (L72)
  4. In part 2, please describe the meaning of “ratio A” and “ratio B”.
  5. Change the form of “P-value” as “P-value”
  6. Check the spelling mistake “inclusion” in L401.
  7. Check the sentence in L425, If from this reference [50], please change the description of the sentence.

Author Response

(The authors gave the same response as above.)

Reviewer 3 Report

The use of plasma amino acids as a "benchmark" should be explained on physiological grounds; plasma AA are usually only 5 % of the body's free AA content which is only about 5% of all the amino acids found in the proteins (bound amino acids) of an organism.

Do the authors really believe that even more refined methods and consequential data will have desired biological and economic outcomes?

The whole discussion is based on digestibility/quantitative data. What about the role of dietary protein quality or has been extensive poultry AA research established that this approach will yield the most useful results?

All in all, this is a useful review for new and veteran amino acid nutrition researchers with many useful citations in poultry (pigs) amino acid  nutrition.

Author Response

(The authors gave the same response as above.)
